# ZnO Nanoparticle-Mediated Seed Priming Induces Biochemical and Antioxidant Changes in Chickpea to Alleviate Fusarium Wilt

**DOI:** 10.3390/jof8070753

**Published:** 2022-07-21

**Authors:** Muhammad Farooq Hussain Munis, Khalid H. Alamer, Ashwaq T. Althobaiti, Asif Kamal, Fiza Liaquat, Urooj Haroon, Junaid Ahmed, Hassan Javed Chaudhary, Houneida Attia

**Affiliations:** 1Department of Plant Sciences, Faculty of Biological Sciences, Quaid-i-Azam University, Islamabad 45320, Pakistan; fmeerani@bs.qau.edu.pk (F.); kamal@bs.qau.edu.pk (A.K.); uharoonsheikh@bs.qau.edu.pk (U.H.); junaidahmed@bs.qau.edu.pk (J.A.); hassaan@qau.edu.pk (H.J.C.); 2Biological Sciences Department, Faculty of Science and Arts, King Abdulaziz University, Rabigh 21911, Saudi Arabia; kalamer@kau.edu.sa; 3Department of Biology, College of Science, Taif University, P.O. Box 11099, Taif 21944, Saudi Arabia; a.t.thobaiti@tu.edu.sa (A.T.A.); hunida.a@tu.edu.sa (H.A.); 4School of Agriculture and Life Science, Seoul National University, 1 Gawanak-gu, Seoul 08826, Korea; fiza.liaquat@bs.qau.edu.pk

**Keywords:** mycosynthesis, nanomaterial, characterization, antifungal activity, SEM, TEM

## Abstract

Chickpea (*Cicer arietinum* L.) is one of the main pulse crops of Pakistan. The yield of chickpea is affected by a variety of biotic and abiotic factors. Due to their environmentally friendly nature, different nanoparticles are being synthesized and applied to economically important crops. In the present study, *Trichoderma harzianum* has been used as a stabilizing and reducing agent for the mycosynthesis of zinc oxide nanoparticles (ZnO NPs). Before their application to control Fusarium wilt of chickpea, synthesized ZnO NPs were characterized. X-ray diffraction (XRD) analysis revealed the average size (13 nm) of ZnO NPs. Scanning electron microscopy (SEM) indicated their spherical structure, and energy dispersive X-ray analysis (EDX) confirmed the oxide formation of ZnO NPs. Transmission electron microscopy (TEM) described the size and shape of nanoparticles, and Fourier transform infrared (FTIR) spectroscopy displayed the presence of reducing and stabilizing chemical compounds (alcohol, carboxylic acid, amines, and alkyl halide). Successfully characterized ZnO NPs exhibited significant mycelial growth inhibition of *Fusarium oxysporum,* in vitro. In a greenhouse pot experiment, the priming of chickpea seeds with ZnO NPs significantly increased the antioxidant activity of germinated plants and they displayed 90% less disease incidence than the control. Seed priming with ZnO NPs helped plants to accumulate higher quantities of sugars, phenol, total proteins, and superoxide dismutase (SOD) to create resistance against wilt pathogen. These nanofungicides were produced in powder form and they can easily be transferred and used in the field to control Fusarium wilt of chickpea.

## 1. Introduction

Chickpea is one of the most valuable pulse crops, worldwide [1]. It provides all three major organic components of food (lipids, proteins, carbohydrates), minerals, and energy. It is a vegetarian source of protein and is widely consumed in the livestock industry [2]. Different biotic and abiotic factors affect the yield of chickpea crop. Ascochyta blight [3] and Fusarium wilt [4] are the main diseases of chickpea. *F. oxysporum* f. sp. *ciceris* is a devastating seedborne and soil fungus and is prevalent in all chickpea growing areas, worldwide [5]. It is the most destructive and challenging infectious fungus as it may hold out in the soil and on crop remnants, as chlamydospores, for up to 6 years [6].

Chickpea yield losses owing to *F. oxysporum* range from 10% to 90% globally and can reach 100% under favorable environmental conditions [7]. *F. oxysporum* can infect chickpea plants at any stage of development [8]. Its chlamydospores penetrate the roots, occupy, and metastasize in the xylem, and cause wilting and eventually death of chickpea plants. It produces specific mycotoxins that cause destructive diseases in a variety of crops. *F. oxysporum* has been reported to cause wilting diseases of chickpea, pea, tomato, sweet potato, and banana [9].

Nowadays, nanotechnology is being used commercially in different fields of agriculture, food, drug delivery, cosmetics, cancer theranostics, and many more [10]. A variety of nanoparticles have been explored to control various plant diseases, specifically fusarium wilt [11]. Nanoscale materials are different from microscale materials in mechanical, chemical, and magneto-optical properties. Along with these unique properties, the surface-to-volume ratios standardize nanoparticles as potential agents in biological applications [12]. In biological science, nanomaterials play a very important role and are used as biologically labeled drugs to control and detect pathogens and identify various proteins [13]. The application of nanoparticles (of 1–100 nanometer size) has emerged as an exciting field and has multiple applications in natural and life sciences.

Currently, many chemical and physical methods are being used for the synthesis of nanoparticles. To apply nanoparticles to different food items, people are working on the environmentally friendly and non-toxic synthesis of nanoparticles. In contrast with conventional fabricated methods, biological methods provide novel ideas to produce nano compounds [14]. Biological methods are preferable due to their safety and efficiency in energy input. Utilization of fungi for the synthesis of nanoparticles is more beneficial, in contrast to other microbes and plants as fungi can produce a greater amount of protein, for the capping and reduction of nanoparticles and these proteins provide longer stability to the nanoparticles. Mycosynthesized nanoparticles have more practical and multiple applications in drugs, nanomedicine, and nano-optoelectronics.

For the synthesis of nanoparticles, different beneficial fungi, with good biocontrol ability are used. Among these fungi, *Trichoderma harzianum* is a widely used fungus and has been reported to be used as biopesticides and biofertilizers [15]. The easy growth conditions of this fungus have increased its applications in the fields of biotechnology and nanotechnology [16]. Genus *Trichoderma* has been reported to possess NADH-dependent enzymes and NADH co-enzymes such as nitrate reductase, which plays a key role in the synthesis and capping of nanoparticles [17]. Zinc oxide nanoparticles (ZnO NPs) have been reported to be used as antimicrobial agents for the growth inhibition of different fungi and bacteria [18].

The current study has been designed to optimize the synthesis of ZnO NPs in *T. harzianum*. These ZnO NPs were characterized and used to control Fusarium wilt of chickpea.

## 2. Materials and Methods

### 2.1. Isolation and Identification of Pathogen

The wilted chickpea plants were collected in zipper bags from the fields of Bhakkar district of Punjab, Pakistan, and brought to the molecular plant pathology lab, Quaid-i-Azam University Islamabad, Pakistan. From the infected roots of chickpea plant, the pathogen was isolated and purified on potato dextrose agar (PDA) media at 25–27 °C. For isolation of the pathogen, roots were washed with distilled water properly and cut into small pieces with a sterilized blade and placed on PDA media. Plates were incubated for five days at 27 °C. Upon mycelial growth, that fungal pathogen was sub-cultured for identification. Microscopy was performed to study the complete morphological characteristics of fungi including hyphae (septate/non-septate) and reproductive structure (sporangia and conidia). The fungus was placed on a clean slide and observed at 40× magnification, under a light microscope [19].

#### 2.1.1. Molecular Identification of Fungus

The fungus isolate was identified, based on the sequence analysis of the 18S ribosomal RNA gene. The CTAB method was used for the extraction of fungal DNA [20]. The quality and quantity of the extracted DNA were checked using Nanodrop^®^. For the amplification of the 18S rRNA gene, forward and reverse primers were used in a polymerase chain reaction (PCR). The PCR mixture was comprised of 0.5 µL of dNTP, 1.5 µL of *Taq* DNA polymerase, 1 µL of genomic DNA, 5 µL of 10× polymerase buffer, and 1 µL of each primer. The PCR reaction was run at 94 °C for 4 min, followed by 35 cycles of 94 °C for 40 s, 58 °C for 40 s, and 72° C for 40 s. The amplified PCR product was sequenced and subjected to BLAST analysis (https://blast.ncbi.nlm.nih.gov/Blast.cgi) accessed on 12 February 2022.

#### 2.1.2. Phylogenetic Analysis of Isolated Fungus

Phylogenetic and evolutionary analyses were performed using MEGA version 7 [21]. The neighbor-joining approach was used to infer the evolutionary relationship of all related fungal strains [22]. The maximum composite likelihood technique was used to compare evolutionary distances [22].

### 2.2. Fugal Culture

*Trichoderma harzianum* (Accession No: FCBP-SF-1277) was obtained from the First Fungal Culture bank of the “Punjab University, Pakistan”, and purified on PDA media. A small portion of mycelia was shaken in broth liquid culture for seven days at 28 °C. The fungal spores were filtered, and the mycelia were washed 2 to 3 times, with distilled water. Clean fungal biomass (35 g) was mixed in 150 mL of sterilized double distilled H_2_O and cultured for a week in an orbital shaking incubator at 150 to 200 rpm and 45 °C. The fungal biomass was then sonicated at room temperature for 30 min, and the pH of the filtrate was monitored.

### 2.3. Preparation of Mycological Zinc Oxide Nanoparticles (ZnO NPs)

In a beaker, purified filtrate and zinc acetate solution (5 mM) were mixed in a 1:1 ratio, to synthesize ZnO NPs. The mixture was shaken at 150 rpm and 40 °C in a shaking incubator for 24 to 48 h. The conformation of reduction of ZnO NPs was assessed by the change in color of the solution. Samples were centrifuged for 15 min at 10,000 rpm. The pallet was collected, washed, and stored at 40 °C overnight. The nanoparticles were subjected to a furnace for two hours at 500 °C.

### 2.4. Characterization of ZnO NPs

Characterization of synthesis ZnO NPs was performed by the following techniques.

#### 2.4.1. UV-Visible Spectroscopic Analysis

For the confirmation of the reduction process of ZnO NPs, the UV–vis spectrum of the solution was evaluated using a UV–vis spectrophotometer (Shimadzu model UV-1601) in the range of 300 to 800 nm.

#### 2.4.2. Fourier Transform Infrared (FTIR) Spectroscopy and Transmission Electron Microscopy (TEM)

By using the KBr pellet method, FTIR spectroscopy was performed to determine the type and nature of various functional groups and structural properties of mycosynthesized ZnO NPs. For this purpose, a scan range of 400 to 4000 cm^1^ was used [23]. A scanning electron microscope (C Joel Jem-1200 EX II. Acc. Voltage 120 kV. MAG-medium) was performed to examine ZnO NPs.

#### 2.4.3. X-ray Diffraction (XRD)

This technique was used to determine the average size and nature of mycosynthesized ZnO NPs. X’Pert High Score software was used to analyze crystallographic characteristics, and the sizes of ZnO NPs were calculated by using the following formula:D = kλ/βcosθ(1)
where D is the average crystalline size perpendicular to the reflecting planes, β is the full width at the half maximum (FWHM), K represents the shape factor, λ indicates the X-ray wavelength, and θ is the diffraction angle.

#### 2.4.4. Scanning Electron Microscopy (SEM) and Energy Dispersive X-ray (EDX) Analysis

The elemental contents and shapes of nanoparticles were determined by sonicating a suspension of nanoparticles in double distilled water for 7 min. A small portion of the sonicated suspension was placed on conductive tape with a double carbon coating and dried under a lamp. For SEM and EDX examination, a (SEM) thermionic emission system was used.

### 2.5. Antifungal Assay

Antifungal activity of ZnO NPs was determined using the poisoned food technique (Liaquat et al., 2021). PDA media containing various concentrations (0.25 μg/mL, 0.50 μg/mL, 0.75 μg/mL, 1.0 μg/mL) of ZnO NPs was used for this purpose. Around 20–25 mL of medium was poured into Petri plates and solidified. Fungal discs (5 to 7 mm) of 6 days old *F. oxysporum* culture were placed in the center of these Petri plates and incubated at 24–28 °C for 7 days. The medium without ZnO NPs suspension served as the control. Antifungal activity of metalaxyl + mancozeb fungicide was also tested at the same concentrations [(0.25 μg/mL (T1), 0.50 μg/mL (T2), 0.75 μg/mL (T3), 1.0 μg/mL (T4)]. Inhibition of mycelial growth was determined by the following formula:Growth Inhibition Percentage = (C − T)/C × 100(2)

T = average mycelial growth in treatments; C = average mycelial growth under control conditions.

### 2.6. Application of ZnO NPs, In Vivo

A pot experiment was carried out to check the effect of ZnO NPs on the growth and disease resistance of chickpea. The experiment was performed in three replicates with a complete randomized block design (CRBD).

The following steps were followed to execute this activity:

#### 2.6.1. Preparation of Fungus Inoculum

Agar plugs were taken from a fresh Petri plate of *F. oxysporum* with sterilized forceps and mixed with autoclaved sorghum seeds, in a conical flask. These inoculum-containing seeds were incubated for 5–6 days at 25–27 °C. For uniform growth, the flask was shaken vigorously, twice a day.

#### 2.6.2. Soil Preparation

For the sowing, germination, and growth of chickpea plants, the soil was prepared by mixing soil, peat moss, and sand in a 1:1:1 ratio and autoclaved after sieving. Pots were filled with approximately 200 g of this soil.

#### 2.6.3. Collection and Surface Sterilization of Seeds

Healthy seeds of chickpea Kabuli variety (Noor 2019) were obtained from (NARC) Agricultural Research Centre Islamabad. The surface of healthy seeds was sterilized for 10 min with 2% sodium hypochlorite solution. After washing with distilled water, the seeds were shade dried.

#### 2.6.4. Seed Priming and Sowing

Based on in vitro mycelial growth inhibition of fungus, a 0.50 μg/mL concentration of ZnO NPs was used for seed priming. The experiment was performed in four treatments. In the first treatment, chickpea seeds were not primed and sown in the normal soil. It served as a control (C). In the second treatment, nano-priming of seeds was performed with a 0.50 µg/mL concentration of ZnO NPs (NPs). In the third treatment, seeds were not primed and sown in fungus-contaminated soil (F). In the fourth treatment, seeds were primed with ZnO NPs and sown in fungus-contaminated soil (F + NPs).

#### 2.6.5. Sowing and Germination of Seeds

For each treatment, three pots were used. In each pot, 10 seeds were sown and placed in a glass house. Based on the emergence of the radicle (2–4 mm length), the germination was recorded after every 24 h. After 10 days of sowing, the germination percentage was determined by using the following equation [24].
GP = (n/N) × 100(3)
where GP = germination percentage; n = number of germinated seeds; N = total number of seeds.

### 2.7. Measurement of Disease Severity

Visual assessment of wilting was observed by following a standard method [25], in which the visual disease scoring was performed by observing the wilting of plants in various treatments. Visual assessment of yellowing was also recorded by following a well-defined previous protocol [26].

### 2.8. Physiological Parameters

After the emergence of true leaves, physiological parameters were studied. The root shoot length of freshly harvested plants was measured by using a measuring tape.

The fresh weight of shoots was calculated by using a sensitive weighing balance. For the calculation of dry weight, the samples were placed in an oven at 70 °C for 24 h and the dry weight of shoot samples was recorded after 72 h.

### 2.9. Biochemical Parameters

After 21 days of sowing, the following biochemical parameters and enzymatic activities were studied.

#### 2.9.1. Estimation of Total Protein, Phenol, Sugar, and Flavonoid Contents

Protein contents and phenol content of the leaves were determined by following standard protocols [27]. For the determination of the total sugar content of chickpea, a previous study was followed [28]. Using the standard spectrophotometer technique, the flavonoid concentration in chickpea leaves was measured.

#### 2.9.2. Enzymatic Activity (SOD, POD and CAT)

Catalase activity was recorded by measuring the decrease in H_2_O_2_ absorbance at 240 nm. For CAT activity analysis, a 3 mL reaction mixture containing 75 mM H_2_O_2_, 100 μL enzyme, and K3PO4 buffer (pH 7.0) was used. One unit of CAT enzyme was expressed in μmol H_2_O_2_ decomposed min^−1^ mg^−1^ protein [5]. SOD was assayed by measuring its ability to inhibit the photochemical reduction of nitro blue tetrazolium [5,29]. For the estimation of SOD enzyme, a 3 mL reaction mixture containing 75 mM NBT, 50 mM phosphate buffer (pH 7.8), 13 mM methionine, 0.9 mL H_2_O, 50 mM sodium carbonate, and 0.1 mL crude extract was used. The reaction was initiated by the addition of 2 mM riboflavin and the absorbance of the reaction was measured at 560 nm. One unit of SOD was expressed in units of min^−1^ mg^−1^ protein POD activity and assessment was determined by a previously described method [5,30]. For POD activity analysis, a 3 mL mixture of pyrogallol (0.05 M) and crude extract (0.5 mL) was used, and the measurement was recorded at 240 nm after adding 1% H_2_O_2_ (*v*/*v*). Finally, 1 O.D. value min^−1^ mg^−1^ protein was considered one unit of enzyme.

### 2.10. Statistical Analyses

The statistical R software was used to analyze the experimental data (ver. 2.14.0). The data in the tables and figures are the averages of three replicates with standard error (SE). A one-way analysis of variance (ANOVA) was used for further analysis. A post hoc test was used to determine the significance of the difference between treated and control plants. At a *p*-value less than 0.05 (≤0.05), the difference was considered significant.

## 3. Results

### 3.1. Isolation and Identification of Fungus

Wilted plants were observed in the field (Figure 1A). On PDA media, the pathogen was isolated (Figure 1B) and its colonies appeared slightly pinkish to whitish, after a week (Figure 1C). Mycelia were possessing fine walls and septa. Micro conidia were aseptate and reniform or oval (Figure 1D). The information of standard conidial and morphological characteristics revealed this pathogen to be *Fusarium oxysporum f.* sp. *ciceris* [20].

#### Molecular Identification of Fungus

The resulting PCR sequence was 100% similar to *Fusarium oxysporum* (Accession No. MT649544). Phylogenetic analysis also showed an evolutionary relationship of our isolate with a specific strain of *F. oxysporum* (Figure 2).

### 3.2. Characterization of ZnO NPs

In this study, ZnO NPs were successfully characterized by the following techniques:

#### 3.2.1. UV-Visible Spectroscopy

Surface plasmon resonance observation at 340 nm wavelength indicated the successful synthesis of ZnO NPs (Figure 3). The color of the cell-free culture changed from yellowish to whitish, indicating the reduction of zinc acetate by secondary metabolites of cell-free filtrate, which leads to the formation of ZnO NPs [31]. Moreover, the difference in the shape of absorbance peaks of zinc oxide and filtrate represented the reduction of zinc acetate salt into ZnO NPs. A similar peak of absorption for zinc oxide at 340 nm has been reported earlier [32].

#### 3.2.2. Fourier Transform Infrared (FTIR) Spectroscopy

FTIR analyses were used for the determination of possible active biomolecules (Figure 4). These biomolecules are accountable for the reduction of Zn ions and capping of ZnO NPs. Obtained FTIR spectra showed absorption peaks at 3379.9 cm^−1^, indicating the NH group of primary aliphatic amines. The peak at 1070.83 cm^−1^ indicated the C-O group of alcohol and strong stretching at 587.01 cm^−1^ depicted the C-I group of halogen. The peaks at 566.79 cm^−1^, 536.18 cm^−1^, 527.10 cm^−1^, and 520.03 cm^−1^ identified the C-Br group of halogen compounds. Another strong stretching peak at 547.07 cm^−1^ displayed the C-Cl group of halogen compounds. The presence of amide groups and organic molecules indicates their role as reducing and capping agents.

#### 3.2.3. X-ray Diffraction (XRD) Structural Analysis

The XRD pattern showed various diffraction peaks (Figure 5), indicating the successful conversion of the precursor salt into ZnO NPs, during the process of calcination. The observed peaks were compared with standard JCPDS data (File No: 000361451). The distinct peaks at 2θ = 31, 34, 36, 47, 56, 62, 67, and 69 were assigned to (100), (002), (101), (102), (110), (103), (112), and (201), indicating the spherical structure of ZnO NPs. Using Debye–Scherer’s equation, the average size of the particle was determined as 13 nm. Biological material resulted in the agglomeration of small-sized nanoparticles.

#### 3.2.4. Scanning Electron Microscopy (SEM) and Energy Dispersive X-ray (EDX) Analysis

SEM micrograph showed spherical-shaped ZnO NPs (Figure 6.) The EDX spectrum confirmed a good percentage of zinc and oxygen signals on ZnO NPs (Figure 7). The elemental characterization of the nanoparticles revealed the presence of zinc (70.70%), oxygen (13.87%), aluminum (7.90%), and carbon (7.52%). The highest peaks of zinc and oxygen confirmed the synthesis of zinc oxide by the metabolites of the filtrate.

#### 3.2.5. Transmission Electron Microscopy (TEM)

TEM can also be used to confirm the primary size and form of nanoparticles [33]. Results of this analysis revealed the spherical shape of ZnO NPs, having an average size of 14 nm (Figure 8). These results also confirmed the findings of XRD and SEM analysis. Size distribution indicated that the ZnO NPs were in the range of 5 nm to 27 nm (Figure 9).

### 3.3. In Vitro Antagonism of F. oxysporum by T. harzianum Mediated ZnO NPs

Different concentrations of ZnO NPs and standard fungicide (metalaxyl + mancozeb) showed variable growth inhibition against *F. oxysporum* (Figure 10). The findings of this study showed that all concentrations of ZnO NPs can exhibit variable growth inhibitions of *F. oxysporum* (Table 1). Among all tested concentrations, 0.50 µg/mL concentration gave the best results (85.2%), followed by 0.25 µg/mL concentration (75.1%). Mycelial growth inhibition at 0.50 µg/mL concentration of NPs has also been reported earlier [34]. In the case of chemical fungicide, 1.0 µg/mL concentration showed the best mycelial inhibition (85.0%). Conclusive studies revealed that lower concentrations of ZnO NPs have great potential to inhibit fungal growth; hence, they are better than chemical fungicides.

### 3.4. Germination Percentage

Seed priming with nanoparticles improved seed germination under both normal and fungal stress conditions (Figure 11). The highest germination percentage (80%) was observed in NPs treatment. In F + NPs treatment, a germination percentage of 70% was observed, which was much better than F treatment.

### 3.5. Disease Analyses

Inoculation of *F. oxysporum* leads to the infection and induction of disease symptoms on all germinated plants (Figure 12). Due to no fungus inoculation, NPs treated and control plants did not show any sign of wilting. Fungus inoculated plants showed severe disease symptoms, while F + NPs-treated plants showed slight wilting symptoms (Figure 13A).

After 21 days of fungal stress (F treatment), lower leaves started turning yellow (Figure 13B). Half of the plants showed yellowing symptoms after 41 days. Nanoparticles-treated plants (NPs treatment) were completely lacking any symptoms, while F + NPs treatment exhibited mild yellowing symptoms.

### 3.6. Disease Control Assay, In Vivo

To control Fusarium wilt of chickpea, a pot experiment was conducted successfully, and following physiological parameters, biochemical parameters, and enzymatic activities were studied.

#### Physiological Parameters

Root and shoot lengths of germinated plants were improved with the application of ZnO NPs (Figure 14A,B). In F treatment, a significant decrease (57%) in the root shoot length was observed, in comparison to the control. F + NPs treatment showed a significantly better root-shoot length than F treatment. Fresh root/shoot ratios of ZnO NPs-treated plants were the greatest among all treatments (Figure 15). The fresh root/shoot ratio of diseased plants (F treatment) was reduced by 20% while an increased fresh root/shoot ratio (35%) was recorded in NPs treatment than in the control treatment. F + NPs treatment also showed an increased fresh root/shoot ratio than F treatment.

### 3.7. Biochemical Parameters

After 21 days of sowing, different biochemical parameters and enzymatic activities were studied to explore the seed priming effect of ZnO NPs. The highest protein contents were observed in ZnO NPs primed plants (Figure 16A). Fungal inoculation (F treatment) resulted in a significant decrease in protein contents, while ZnO NPs improved protein contents in NPs + F treatment.

Plant phenolics are naturally occurring products in the plant body. The highest phenol contents were recorded in NPs-primed plants under fungal stress (Figure 16B). It positively correlates with the reduction in wilting of the plants.

The highest sugar content was recorded in ZnO NPs primed plants, followed by primed plants under fungal stress conditions (Figure 16C). The lowest sugar contents were recorded in fungal stressed plants, as compared to control.

The highest flavonoid contents were found in F + NPS treatment, followed by ZnO NPs-treated plants and control plants (Figure 16D). This increase in flavonoid content in chickpea plants helps to mitigate the effects of both biotic and abiotic stresses.

### 3.8. Enzymatic Activities

Both NPs treatment and fungus inoculation increased peroxidase activity, and the maximum enzymatic activity was observed under F + NPs treatment (Figure 17A). Catalase (CAT) enzyme showed the same trend (Figure 17B), and the maximum superoxide dismutase (SOD) activity was observed in NPs-treated plants, followed by F + NPs treatment (Figure 17C). Increased SOD in plants treated with ZnO NPs could be one technique for removing excess ROS and limiting pathogen invasion [5].

## 4. Discussion

*F. oxysporum* is a disease-causing pathogen of many economically important crops. In this study, *F. oxysporum* has been reported to cause wilting of chickpea in different chickpea-growing regions of the world. It has been reported as a wilting pathogen of chickpea in different parts of the world [7]. Various practices are being used for the control of *F. oxysporum* [35], but the ultimate control of this fungus is very difficult due to its persistent and diverse nature. *F. oxysporum* is a soil and seed-borne fungus, so the application of fungicide in soil is very difficult and non-sustainable. The application of various agrochemicals leads to environmental and health issues which are encountered with their excessive use [36]. In this study, a species of *Trichoderma* was used for the synthesis of ZnO NPs. *Trichoderma* is a plant growth-promoting fungus, and it is utilized as a biocontrol agent. Various arbuscular mycorrhizal fungi are used as biocontrol agents for plant diseases [37]. *T. harzianum* isolate Tr-3 has also been reported as a powerful biocontrol agent against *B. cineria* and also shows remarkable mycelial growth inhibition of *B. cineria* in tomato [38]. It also has the ability to combat a wide range of phytopathogens by releasing antibiotics and establishing plant systemic resistance. It is a preferable fungus for the synthesis of various metal oxides [39].

Fungus-based ZnO NPs are widely used as antifungal agents against various fungi, i.e., *A. niger**, A. fumigatus*, and *A. aculeatus* These fungi-based ZnO NPs are environmentally friendly and non-hazardous in nature. Their use in crops and fruits is encouraged. They also possess antioxidant, antibacterial, and antifungal properties [40], and also have the potential to remediate heavy metal stress [41]. They are effective inhibitors of mycelial growth because they break microbe sheaths and create reactive oxygen species (ROS). The NPs migrate via nanometer pores after rupturing the membrane of the microbial cell [42].

In this study, the characterization of ZnO NPs showed their stable nature and suitable size. FTIR spectroscopy describes the availability of various functional groups during redox the reaction of NPs synthesis [43]. In this study, the characterization of ZnO NPs showed their stable crystalline nature, and the small size (13 nm) of mycosynthesized NPs was revealed. According to studies, the smaller size of NPs has high antibacterial effects [44]. SEM examination is particularly beneficial for determining nanoparticle morphology [45]. *T. harzianum*-mediated zinc oxide nanoparticles had a comparable spherical form as *Agaricus bisporus* (button mushroom)-mediated zinc oxide nanoparticles. Very prominent peaks of zinc (Zn) and oxygen (O) were identified in the EDX spectrum of the current investigation, which describes and validates the pure oxide form of ZnO NPs [46].

It was also reported that ZnO NPs have fungicidal effects against a variety of pathogens, i.e., *Rhizopus stolonifier* [47]. In this investigation, mycosynthesized ZnO NPs inhibited the growth of *F. oxysporum*, both in vitro and in vivo. In our study, maximum mycelial growth inhibition was observed at 0.50 μg/mL concentration of ZnO NPs. In previous studies, higher concentrations have also been observed to show poor antifungal activities [48]. At higher concentrations, nanoparticles agglomerate with each other and form bulk particles. It was also reported that a higher concentration of zinc oxide has toxic effects on the emergence and seedling growth of chickpea.

In our study, nano-priming induced a positive effect on the growth of chickpea plants. Similar improvements in the morphology, physiology, biochemistry, and enzymatic activities of chickpea have been reported earlier [49]. In addition to this, foliar application of ZnO NPs has also been reported to improve the physiology of plants. Application of ZnO NPs improved the photosynthetic rate and fresh and dry weight of roots [50]. Seed priming with NPs suspension also plays a key role in the suppression of disease, caused by *F. oxysporum* [51]. ZnO NPs not only display disease suppression properties but also have the ability to increase the shelf life of vegetables and fruits [47]. ZnO NPs trigger the upregulation of potential antioxidant metabolism [52]. Our findings revealed an overall increase in enzymatic activities. Similar results have also been reported by the application of chitosan nanoparticles in wheat [52]. The activation of defense enzymes such as PO, SOD, CAT, phenol, and flavonoids is known to boost host tolerance under biotic stress, particularly fungal stress [5].

## 5. Conclusions

ZnO NPs are very effective agents to control *Fusarium* wilt of chickpea. They are non-toxic and environmentally friendly, and they can be easily used for the priming of chickpea seeds. *T. harzianum* mycotoxins can effectively cover and stabilize ZnO NPs, and these NPs exhibit strong antifungal potential and can improve the physiology and biochemistry of plants. These NPs are produced in powder form, and they can be easily transported to the field for their large-scale application.

## Figures and Tables

**Figure 1 jof-08-00753-f001:**
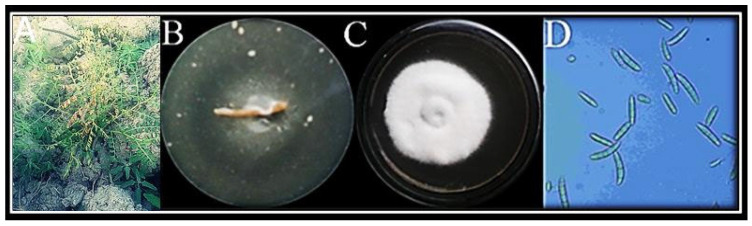
Wilted plants were observed in the field (**A**). Fungus was isolated from roots (**B**) and sub-cultured on PDA media (**C**). Microscopic image of isolated fungus was obtained at 40× magnification (**D**).

**Figure 2 jof-08-00753-f002:**
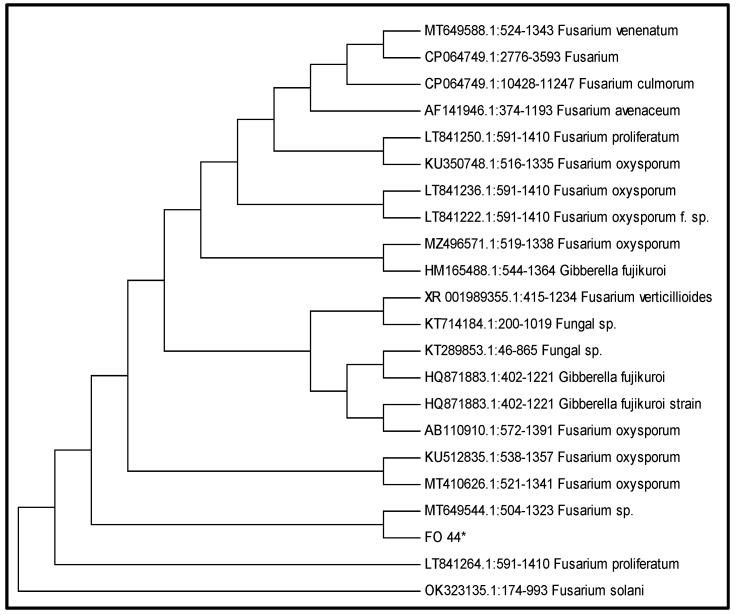
Evolutionary relationship of isolated *Fusarium oxysporum* strain FO_44, with 21 closely related GenBank sequences.

**Figure 3 jof-08-00753-f003:**
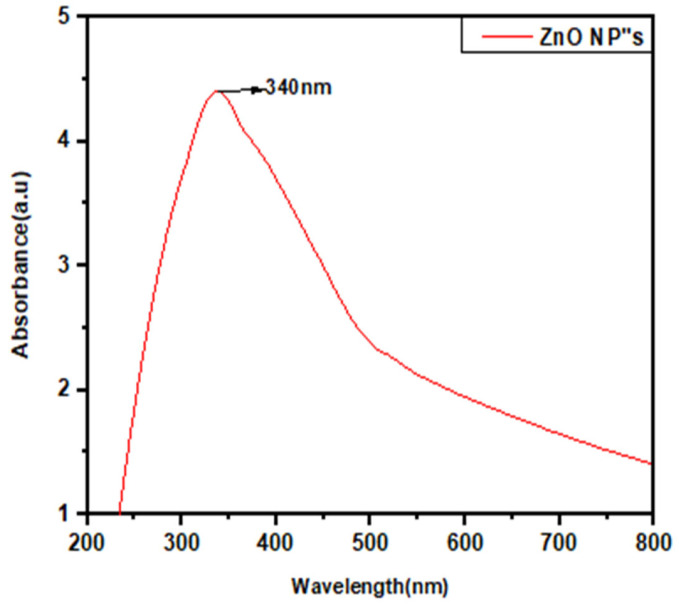
UV–visible spectrum of ZnO NPs.

**Figure 4 jof-08-00753-f004:**
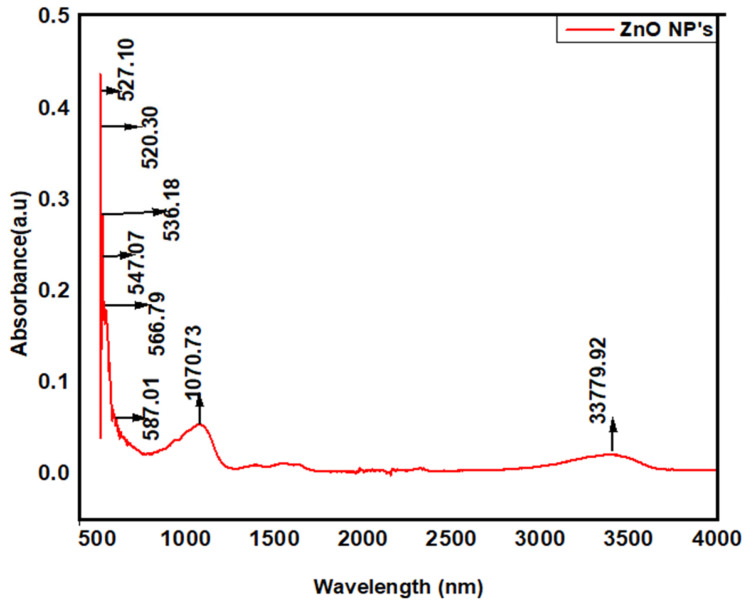
FTIR spectrum of ZnO NPs.

**Figure 5 jof-08-00753-f005:**
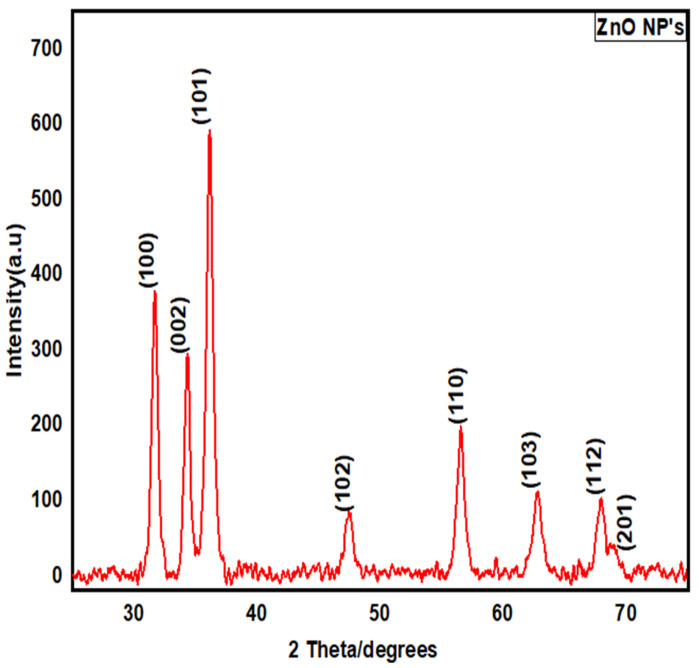
XRD spectrum of mycosynthesized ZnO NPs.

**Figure 6 jof-08-00753-f006:**
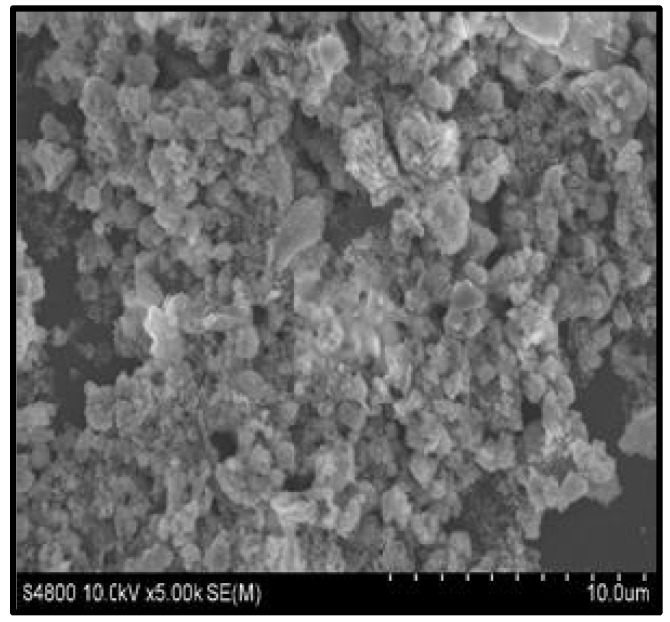
SEM micrographic image of mycosynthesized ZnO NPs.

**Figure 7 jof-08-00753-f007:**
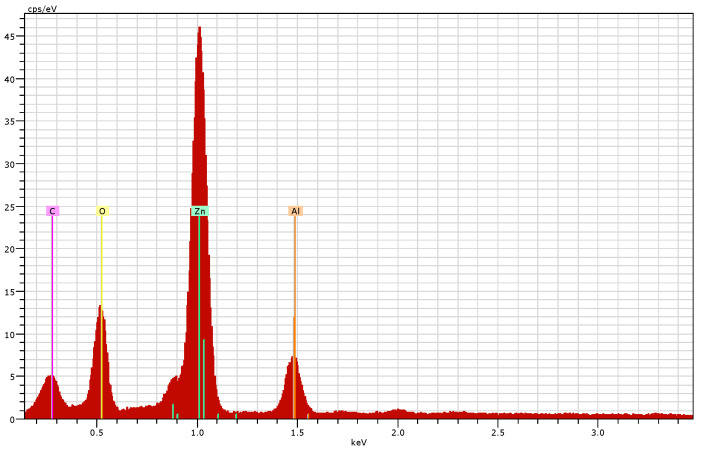
EDX spectrum of mycosynthesized ZnO NPs.

**Figure 8 jof-08-00753-f008:**
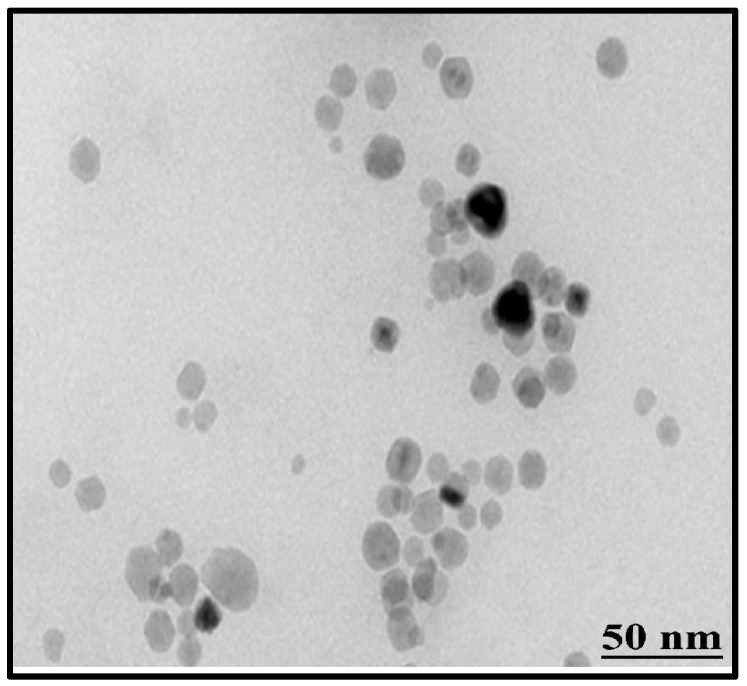
TEM micrographic image of mycosynthesized ZnO NPs.

**Figure 9 jof-08-00753-f009:**
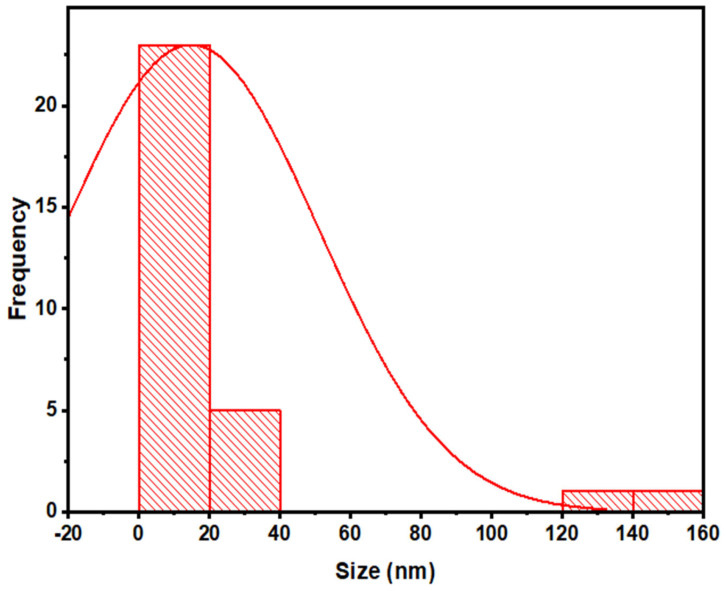
TEM micrograph showing size distribution of mycosynthesized ZnO NPs.

**Figure 10 jof-08-00753-f010:**
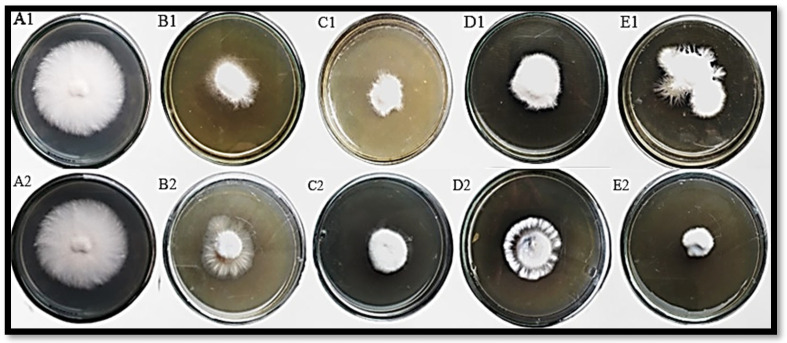
Antagonism of *F. oxysporum* under various concentration of ZnO NPs and chemical fungicides, in vitro. (**A1**) Control, (**B1**) 0.25 μg/mL ZnO NPs, (**C1**) 0.50 μg/mL ZnO NPs, (**D1**) 0.75 μg/mL ZnO NPs, (**E1**) 1.0 μg/mL ZnO NPs, (**A2**) control, (**B2**) 0.25 μg/mlL metalaxyl + mancozeb fungicide, (**C2**) 0.50 μg/mL metalaxyl + mancozeb fungicide, (**D2**) 0.75 μg/mL metalaxyl + mancozeb fungicide, (**E2**) 1.0 μg/mL metalaxyl + mancozeb fungicide.

**Figure 11 jof-08-00753-f011:**
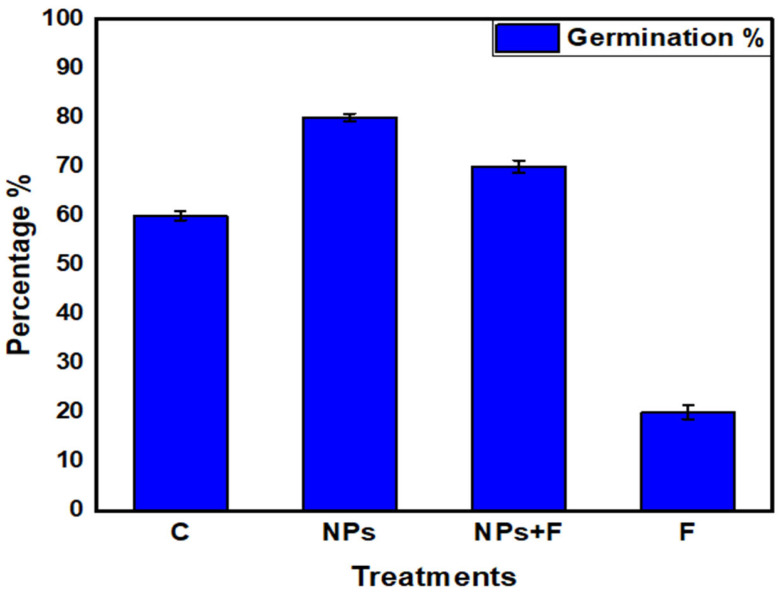
Germination percentage of chickpea seeds under different growth conditions.

**Figure 12 jof-08-00753-f012:**
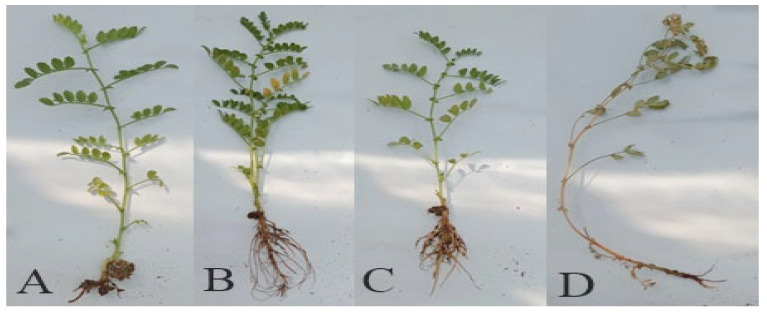
Visual assessment of wilting and yellowing. Control (**A**), seeds primed with NPs (**B**), Seeds primed with 500 mg/L nanoparticles and sown in fungus-contaminated soil (**C**). Seeds sown in fungus-contaminated soil (**D**).

**Figure 13 jof-08-00753-f013:**
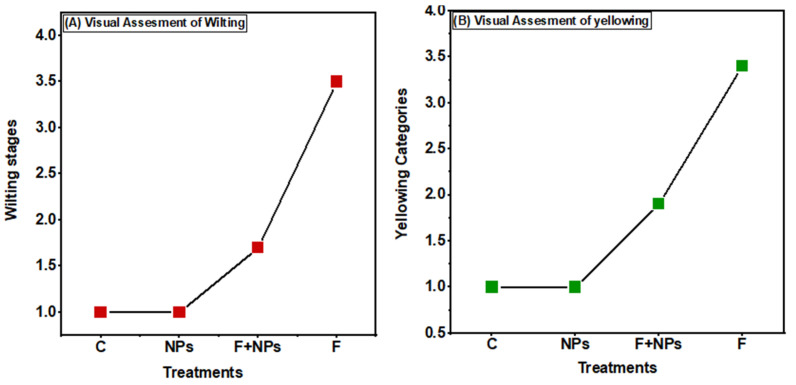
Visual assessment of wilting (**A**) and yellowing (**B**). C = control (without fungus and without priming), NPs = seeds primed with 0.5 µg/mL NPs, F + NPs = seeds primed with 0.5 µg/mL and sown in fungus-contaminated soil, F = seeds sown in fungus-contaminated soil.

**Figure 14 jof-08-00753-f014:**
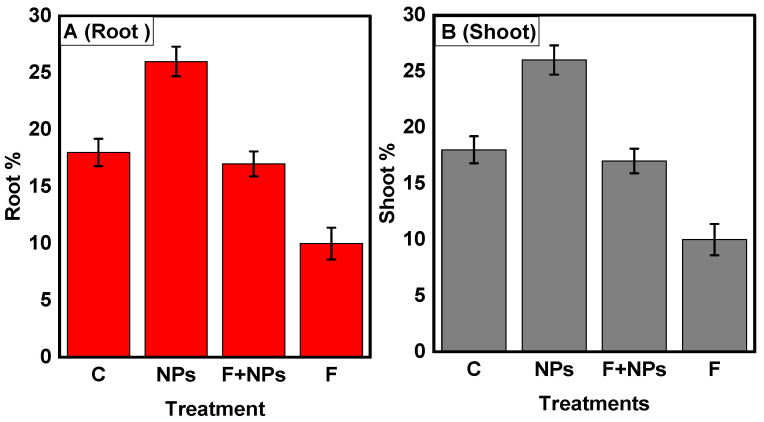
Root length (**A**) and shoot length (**B**) of chickpea under different growth conditions. C = control (without fungus), NPs = seeds primed with 0.5 µg/mL ZnO NPs, NPs + F = seeds primed with 0.5 μg/mL ZnO NPs and sown in fungus-inoculated soil, F = seeds sown in fungus-inoculated soil.

**Figure 15 jof-08-00753-f015:**
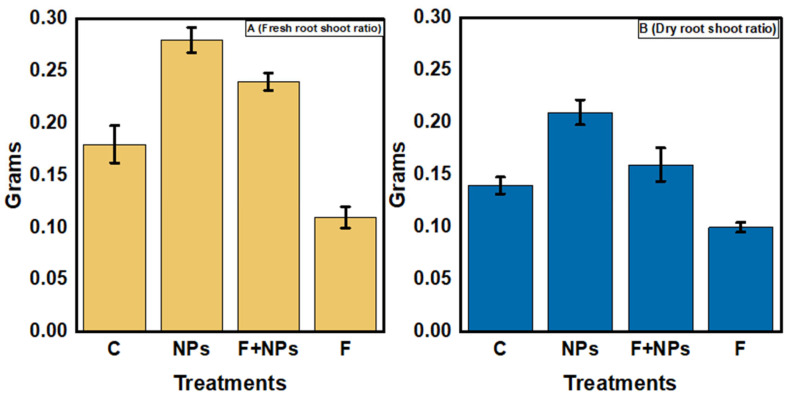
Fresh root/shoot ratios (**A**) and dry root/shoot ratios (**B**) of chickpea under different growth conditions. C = control (without fungus), NPs = seeds primed with 0.5 µg/mL ZnO NPs, NPs + F = seeds primed with 0.5 μg/mL ZnO NPs and sown in fungus inoculated soil, F = seeds sown in fungus inoculated soil.

**Figure 16 jof-08-00753-f016:**
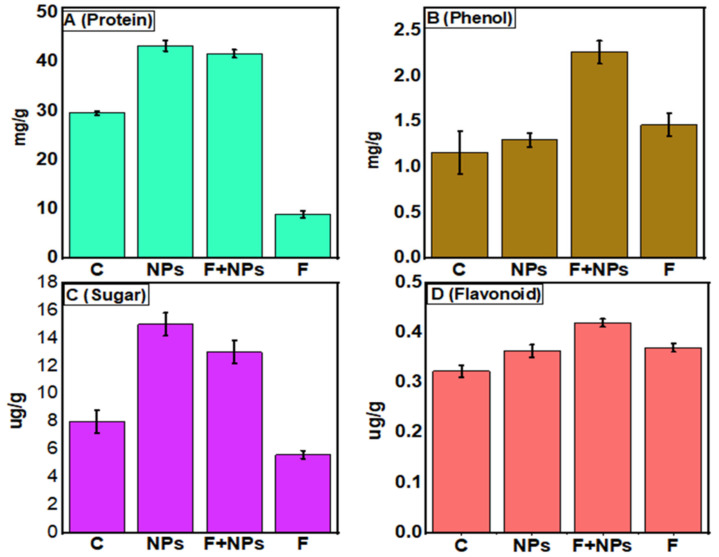
Estimation of different biochemical parameters including Protein (**A**), Phenol (**B**), Sugar (**C**), and Flavonoid (**D**) of chickpea under different growth conditions. C = control (without fungus), NPs = seeds primed with 0.5 µg/mL ZnO NPs, NPs + F = seeds primed with 0.5 ug/mL ZnO NPs and sown in fungus inoculated soil, F = seeds sown in fungus inoculated soil.

**Figure 17 jof-08-00753-f017:**
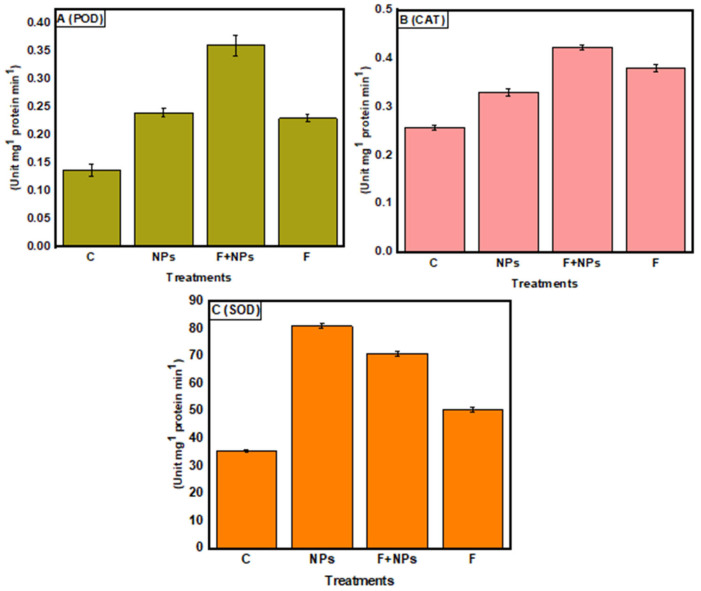
Peroxidase (POD) activity (**A**), catalase activity (**B**) and superoxide dismutase (SOD) activity (**C**) of chickpea under different growth conditions. C = control (without fungus), NPs = seeds primed with 0.5 µg/mL ZnO NPs, NPs + F = seeds primed with 0.5 μg/mL ZnO NPs and sown in fungus inoculated soil, F = seeds sown in fungus inoculated soil.

**Table 1 jof-08-00753-t001:** Percentage growth inhibition of *F. oxysporum* under various concentrations of ZnO NPs and metalaxyl + mancozeb fungicide.

Concentration	ZnO NPs % Inhibition	Metalaxyl + Mancozeb Fungicide % Inhibition
0.25 μg/mL	78.0 ± 0.5	64.1 ± 0.5
0.5 μg/mL	85.2 ± 0.5	66.3 ± 0.0
0.75 μg/mL	75.1 ± 0.5	83.2 ± 0.5
1.0 μg/mL	72.0 ± 0.5	85.0 ± 0.5
Control	95.0 ± 0.5	95.5.0 ± 0.5

## Data Availability

The datasets generated during and/or analyzed during the current study are available from the corresponding author on reasonable request.

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
