# Peer review of "ZnO Nanoparticle-Mediated Seed Priming Induces Biochemical and Antioxidant Changes in Chickpea to Alleviate Fusarium Wilt"

_jof, 2022, doi:10.3390/jof8070753_

Round 1
Reviewer 1 Report
This review is concerning a research work entitled “ ZnO nanoparticle-mediated seed priming induces biochemical and antioxidant changes in chickpea to alleviate Fusarium wilt” by Farhana, et al.
several points have to be precised and a major revision is requested.
1- Although I am not English native speaker, this text contains too many errors and an extensive reading by English-native reader is necessary for a suitable publication in this international journal. I put in the minor points at the end some of these, but the whole text has to be considered.
2- Please put all latin names in italics throughout the text (it is wrong in quite many places, check them one by one); put also the names of the author(s) of all taxa cited the first time they appear in the text: use international Plant Names Index (IPNI) https://www.ipni.org/) for plants, for bacteria use https://lpsn.dsmz.de/ or equivalent, for the different strains cited in the text you can provide the reference of the first time this strain appeared in the literature; for fungi use the International Code of Nomenclature for algae, fungi, and plants (https://www.iapt-taxon.org/nomen/main.php) or equivalent.
3- Keyword should be different than in title
4- Make sure that all scientific names in the References list are italics.
5- Please add the DOI for ALL the References
6- All tables must be self-explanatory.
7- From where you get Trichoderma? Are you isolated it ? how you identified it?
8- In all Material as well as Results you need to revise the title of your work for example
2.2 you can write main title 2.2. Identification of the pathogen after that subtitle 2.2.1 Microscopic Identification, 2.2.2 Molecular Identification of Fungus
2.7. could be 2.7. Characterization of ZnO NPs and subtitle 2.7.1. UV-visible spectroscopic analysis ……2.7.2. Fourier transform infrared (FTIR) spectroscopy and transmission electron microscopy 130 (TEM) … etc
The same could be in rest of the experiment as well as in Results
9- How many replicate you used for enzyme activities
10- You should write in brief how you determined the enzymes not only the references
11- When you isolated the pathogen only you found one isolates and how you know this is pathogenic isolate?
12- Fig 1 A not clear
13- Fig 11 you do not need to write legend for this fig
14- Discussion part should be improve with more details
15- In part 2.23 is this parameters Physiological study? I do not think
16- Some time author write the refernces within the text number some time write the name you should fellow the Journal style
17- Here some new refernces you can used for improve your discussion
https://doi.org/10.1007/s10343-022-00686-3
https://doi.org/10.1016/j.plaphy.2016.08.022
https://doi.org/10.1016/j.plaphy.2018.12.005
DOI: 10.2478/fhort-2019-0025
https://doi.org/10.3390/agronomy12030671
https://doi.org/10.3390/jof8030304
Author Response
Reviewer 1.
This review is concerning a research work entitled “ZnO nanoparticle-mediated seed priming induces biochemical and antioxidant changes in chickpea to alleviate Fusarium wilt” by Farhana, et al.
Several points have to be precised and a major revision is requested.
- Although I am not English native speaker, this text contains too many errors and an extensive reading by English-native reader is necessary for a suitable publication in this international journal. I put in the minor points at the end some of these, but the whole text has to be considered.
Answer: All changes have been incorporated and the manuscript has been checked for errors, again.
- Please put all latin names in italics throughout the text (it is wrong in quite many places, check them one by one); put also the names of the author(s) of all taxa cited the first time they appear in the text: use international Plant Names Index (IPNI) https://www.ipni.org/) for plants, for bacteria use https://lpsn.dsmz.de/ or equivalent, for the different strains cited in the text you can provide the reference of the first time this strain appeared in the literature; for fungi use the International Code of Nomenclature for algae, fungi, and plants (https://www.iapt-taxon.org/nomen/main.php) or equivalent.
Answer: All the suggested changes have been incorporated. Whole document is cross checked.
- Keyword should be different than in title
Answer: All the suggested changes have been incorporated. Keywords have been changed. (Line no 35).
- Make sure that all scientific names in the References list are italics.
Answer: All the suggested changes have been incorporated.
5- Please add the DOI for ALL the References
Answer: DOI has been added.
6- All tables must be self-explanatory.
Answer: All the suggested changes have been incorporated. (Line no 341).
7- From where you get Trichoderma? Are you isolated it? how you identified it?
Answer: We did not isolate Trichoderma. It has been taken from First Fungal Culture Bank of Pakistan Punjab University in Identified form.
- In all Material as well as Results you need to revise the title of your work for example 2.2 you can write main title 2.2. Identification of the pathogen after that subtitle 2.2.1 Microscopic Identification, 2.2.2 Molecular Identification of Fungus 2.7. could be 2.7. Characterization of ZnO NPs and subtitle 2.7.1. UV-visible spectroscopic analysis ……2.7.2. Fourier transform infrared (FTIR) spectroscopy and transmission electron microscopy 130 (TEM) … etc. The same could be in rest of the experiment as well as in Results
Answer:. Suggested changes have been incorporated.
- How many replicates you used for enzyme activities
Answer: Three replicates have been used.
- You should write in brief how you determined the enzymes not only the references.
Answer: Brief Methodology has been added. (Line no 232, 236 and 241).
- When you isolated the pathogen only you found one isolates and how you know this is pathogenic isolate?
Answer: Pathogen was isolated during survey in Bhakkar district of Punjab, Pakistan in the month of November. In addition to Fusarium pathogen we isolated bacterial isolates but we go for fungal pathogen as it was already reported in literature that it is one of most devastating pathogens of chickpea.
12 Fig 1 A not clear
Answer: Clear figure has been added. (Line no 262).
- Fig 11 you do not need to write legend for this fig
Answer: Legend has removed for Fig 11. (Line no 353).
- Discussion part should be improve with more details.
Answer: Discussion has been improved as per suggestion.
- In part 2.23 is this parameters Physiological study? I do not think
Answer: Thanks for highlighting mistake. Correction has been incorporated.
- Some time author write the refernces within the text number some time write the name you should fellow the Journal style.
Answer: Same style has been followed.
- Here some new refernces you can used for improve your discussion
https://doi.org/10.1007/s10343-022-00686-3
https://doi.org/10.1016/j.plaphy.2016.08.022
https://doi.org/10.1016/j.plaphy.2018.12.005
DOI: 10.2478/fhort-2019-0025
https://doi.org/10.3390/agronomy12030671
https://doi.org/10.3390/jof8030304
Answer: All suggested journals have been cited in the text.
https://doi.org/10.1007/s10343-022-00686-3 (line no 435)
https://doi.org/10.1016/j.plaphy.2016.08.022 (line no 443)
https://doi.org/10.1016/j.plaphy.2018.12.005 (line no 467)
DOI: 10.2478/fhort-2019-0025 (line no 457 and 472)
https://doi.org/10.3390/agronomy12030671 (line no 435)
https://doi.org/10.3390/jof8030304 (line no 477)

Reviewer 2 Report
See the comments in attached file.

Author Response
Reviewer 2
See the comments in attached file.
Answer: All the suggestions have been incorporated into the manuscripts and comments have been addressed in the PDF file.

Reviewer 3 Report
Dear Editor,
the manuscript "ZnO nanoparticle-mediated seed priming induces biochemical and antioxidant changes in chickpea to alleviate Fusarium wilt" by Farhana and colleagues provides information on the use of ZnO nanoparticles in order to alleviate Fusarium wilt in chickpea.
Overall the manuscript is not very good structured. Some figures are not correctly represented and should be improved (legend and interpretation). Figures in the text are not in brackets. Also, the author should improve and correct technical/professional vocabulary. The scientific names in almost entire article are not italicized.
Furthermore, the manuscript several mistakes which should be double checked. Specific comments follow:
· The name of the first author seems to be incomplete
Introduction
· line 43: Please delete the free space
· lines 51-52: Please use more recent literature and add more citations in that paragraph. You can find more recently articles about a variety of nanoparticles used in controlling pathogens such as Fusarium oxysporum.
Please use the following citations:
· Chittarath, K.; Nguyen, C.H.; Bailey, W.C.; Zheng, S.-J.; Mostert, D.; Viljoen, A.; Tazuba, A.F.; Ocimati, W.; Kearsley, E.; Chi, T.Y.; Tho, N.T.; Hung, N.T.; Dita, M.; Shah, T.; Karanja, M.; Mahuku, G.; Blomme, G. Geographical Distribution and Genetic Diversity of the Banana Fusarium Wilt Fungus in Laos and Vietnam. J. Fungi 2022, 8, 46. https://doi.org/10.3390/jof8010046
· Lipșa, F.-D.; Ursu, E.-L.; Ursu, C.; Ulea, E.; Cazacu, A. Evaluation of the Antifungal Activity of Gold–Chitosan and Carbon Nanoparticles on Fusarium oxysporum. Agronomy 2020, 10, 1143. https://doi.org/10.3390/agronomy10081143
· line 76: The scientific names of Trichoderma is not italicized.
Material and methods
The methodology, materials and equipment used in this study and described in point 2 is not adequately presented.
· lines 85-87: For example, point 2.1. is entitled Isolation and Identification of Pathogen, but here it is only shown that it is isolated on PDA medium, but the temperature varies between 25 and 27 degrees Celsius (which is the temperature at which growth of F oxysporum species occurred on the medium). How was the sample taken? Under what conditions was it transported? What was the sterilisation temperature of the medium? Was the medium dehydrated? The information needs to be completed...
· Points 2.2, 2.3 should be included in point 2.1.
· lines 91-92: The fungus was placed on clean slide and observed on at 40× magnification under 91 electron microscope.
For 40x magnification power you don't need an electron microscope but an optical one ...please specify model and manufacturer
· lines 124-125: Points 2.8,-2.11 should be included in point 2.7. Information contained in point 2.7 make sense only with the infos from 2.8-2.11
· lines 163-167: idem - 2.13 should contain infos from2.14-2.18
· idem for 2.19 - should contain infos from 2.20-2.21
· idem for 2.22
Results
· lines 236-238: please rephrase and insert the figures in brackets (please check the entire text) Wilted plants were observed in the field Figure 1A On PDA media, the pathogen was 236 isolated Figure 1B and its colonies appeared slightly pinkish at the center while their edges 237 were white in color Figure 1C.
· line 238-239: please delete Macroconidia were of short to medium length (PLEASE INSERT VALUES) ad (please correct …AND) these were falciform to nearly straight, showing fine walls and septa
· line 241: ciceris and not Ciceris (not capitalized C)
· line 243: Figure 1 point (D)…please mention the conidia and not the scale bar
· lines 252-253: Information contained in point 3.3 make sense only with the information from 3.4-3.8.
· Please rephrase also for 3.10…..
· Please rephrase also for 3.18…..
· lines 308-310: are the data statistical significant???? I didn’t found any information for any data in the manuscript. The best inhibition results are not statistical relevant if the authors are not providing the necessary statistical data.
Findings of this study showed that all concentrations of ZnO NPs can inhibit the growth of F. oxysporum but 0.50 μg/mL concentration gave the best results (85.2%), followed by 0.25 309 μg/mL concentration (75.1%).
· Table 1 (line 321) is not fully explicit. Which is the unit of measurement used for ZNO NPS and fungicide?
Discussion
The information presented in the Discussion is very limited and does not cover all the results presented. This section needs to be completely rethought to highlight the novelty of the paper.
Conclusion
Should be rewritten because the information presented are spare and many results are not presented.
Author Response
Answers to reviewer’s comments
Reviewer 3
This review is concerning a research work entitled “ZnO nanoparticle-mediated seed priming induces biochemical and antioxidant changes in chickpea to alleviate Fusarium wilt” by Farhana, et al.
- Overall, the manuscript is not very good structured. Some figures are not correctly represented and should be improved (legend and interpretation). Figures in the text are not in brackets. Also, the author should improve and correct technical/professional vocabulary. The scientific names in almost entire article are not italicized.
Answer: All changes have been incorporated.
Furthermore, the manuscript several mistakes which should be double checked. Specific comments follow:
. The name of the first author seems to be incomplete
Answer: All changes have been incorporated.
Introduction
- Line 43: Please delete the free space; Lines 51-52: Please use more recent literature and add more citations in that paragraph. You can find more recently articles about a variety of nanoparticles used in controlling pathogens such as Fusariumoxysporum.
Answer All changes have been incorporated. (Line no 43)
- Please use the following citations:
Chittarath, K.; Nguyen, C.H.; Bailey, W.C.; Zheng, S.-J.; Mostert, D.; Viljoen, A.; Tazuba, A.F.; Ocimati, W.; Kearsley, E.; Chi, T.Y.; Tho, N.T.; Hung, N.T.; Dita, M.; Shah, T.; Karanja, M.; Mahuku, G.; Blomme, G. Geographical Distribution and Genetic Diversity of the Banana Fusarium Wilt Fungus in Laos and Vietnam. J. Fungi 2022, 8, 46. https://doi.org/10.3390/jof8010046
Lipșa, F.-D.; Ursu, E.-L.; Ursu, C.; Ulea, E.; Cazacu, A. Evaluation of the Antifungal Activity of Gold–Chitosan and Carbon Nanoparticles on Fusarium oxysporum. Agronomy 2020, 10, 1143. https://doi.org/10.3390/agronomy10081143
Answer: Above mentioned articles have been cited (Line 53 and Line 56)
- line 76: The scientific name of Trichodermais not italicized.
Answer: Scientific names are italicized. (Line no 75)
Material and methods
- The methodology, materials and equipment used in this study and described in point 2 is not adequately presented. Lines 85-87: For example, point 2.1. is entitled Isolation and Identification of Pathogen, but here it is only shown that it is isolated on PDA medium, but the temperature varies between 25 and 27 degrees Celsius (which is the temperature at which growth of F oxysporum species occurred on the medium). How was the sample taken? Under what conditions was it transported? What was the sterilisation temperature of the medium? Was the medium dehydrated? The information needs to be completed.
Answer: All changes have been incorporated. (Line no 88 to 94)
- Points 2.2, 2.3 should be included in point 2.1.
Answer: All changes have been incorporated. (Line no 96)
- lines 91-92: The fungus was placed on clean slide and observed on at 40× magnification under electron microscope. For 40× magnification power you don't need an electron microscope but an optical one ...please specify model and manufacturer
Answer: Highlighted information has been corrected and light microscope has been mentioned in Line 99.
- lines124-125: Points 2.8,-2.11 should be included in point 2.7. Information contained in point 2.7 make sense only with the infos from 2.8-2.11
Answer: As per instruction, points 2.2 and 2.3 have been merged in 2.1.
- lines 91-92: The fungus was placed on clean slide and observed on at 40× magnification under electron microscope. For 40x magnification power you don't need an electron microscope but an optical one ...please specify model and manufacturer
Answer: It was a light microscope. Word Electron microscope was mistakenly written.
- lines124-125: Points 2.8,-2.11 should be included in point 2.7. Information contained in point 2.7 make sense only with the infos from 2.8-2.11
Answer: As per instruction, Points 2.8,-2.11 has been included in point 2.7..
- lines 163-167: idem - 2.13 should contain infos from 2.14-2.18
Answer: As per instruction, Points 2.14,-2.18 have been included in point 2.13.
- idem for 2.19 - should contain infos from 2.20-2.21
Answer: As per instruction, Points 2.19 is now containing infos from infos from 2.20-2.21.
Results
- lines 236-238: please rephrase and insert the figures in brackets (please check the entire text) Wilted plants were observed in the field Figure 1A On PDA media, the pathogen was 236 isolated Figure 1B and its colonies appeared slightly pinkish at the center while their edges 237 were white in color Figure 1C.
Answer: Changes has been incorporated.
- line 238-239: please delete Macroconidia were of short to medium length(PLEASE INSERT VALUES) (please correct …AND) these were falciform to nearly straight, showing fine walls and septa.
Answer: Changes has been incorporated. (Line no 54 to 59)
- line 241: ciceris and not Ciceris (not capitalized C)
Answer: Changes has been incorporated. (Line no 59)
- line 243: Figure 1 point (D)…please mention the conidia and not the scale bar
Answer: Changes has been incorporated. (Line no 262)
- lines 252-253: Information contained in point 3.3 make sense only with the information from 3.4-3.8.
Answer: Changes has been incorporated.
- Please rephrase also for 3.10…..
Answer: Changes has been incorporated. (Line no 345 to 346)
- Please rephrase also for 3.18…..(Line no 304 to 305)
Answer: Changes has been incorporated.
- lines 308-310: are the data statistical significant???? I didn’t found any information for any data in the manuscript. The best inhibition results are not statistical relevant if the authors are not providing the necessary statistical data. Findings of this study showed that all concentrations of ZnO NPs can inhibit the growth of F. oxysporum but 0.50 μg/mL concentration gave the best results (85.2%), followed by 0.25 309 μg/mL concentration (75.1%).
Answer: Changes has been incorporated.
- Table 1 (line 321)is not fully Which is the unit of measurement used for ZNO NPS and fungicide?
Answer: Changes has been incorporated; measurement has been taken in percentage. (Line no 341 to 342)
Discussion
The information presented in the Discussion is very limited and does not cover all the results presented. This section needs to be completely rethought to highlight the novelty of the paper.
Answer: Changes has been incorporated in whole discussion (Line 451 to 494)
Conclusion
Should be rewritten because the information presented are spare and many results are not presented.
Answer: Changes has been incorporated in conclusion.

Round 2
Reviewer 1 Report
I see authors covered my previous comments
Reviewer 3 Report
I agree with this form